# CRISPR/Cas9-Induced Knockout of *Sting* Increases Susceptibility of Zebrafish to Bacterial Infection

**DOI:** 10.3390/biom13020324

**Published:** 2023-02-08

**Authors:** Sarithaa Sellaththurai, Sumi Jung, Myoung-Jin Kim, Kishanthini Nadarajapillai, Subothini Ganeshalingam, Joon Bum Jeong, Jehee Lee

**Affiliations:** 1Department of Marine Life Sciences, Jeju National University, Jeju 63243, Republic of Korea; 2Marine Science Institute, Jeju National University, Jeju 63333, Republic of Korea; 3Nakdonggang National Institute of Biological Resources, Sangju 37242, Republic of Korea; 4Fish Vaccine Research Center & Center for Genomic Selection in Korean Aquaculture, Jeju National University, Jeju 63243, Republic of Korea

**Keywords:** zebrafish, CRISPR/Cas9, stimulator of interferon gene, *Edwardsiella piscicida*

## Abstract

Stimulator of interferon genes (STING) is an adapter protein that is activated when cyclic dinucleotides (CDNs) are present. CDNs originate from the cytosolic DNA of both pathogens and hosts. STING activation promotes efficient immune responses against viral infections; however, its impact in bacterial infections is unclear. In this study, we investigated the role of Sting in bacterial infections by successfully creating a *sting*-deficient (*sting^(−/−)^* with a 4-bp deletion) knockout zebrafish model using CRISPR/Cas9. The transcriptional modulation of genes downstream of cGAS (cyclic GMP-AMP synthase)-Sting pathway-related genes was analyzed in seven-day-old wild-type (WT) and *sting^(−/−)^* embryos, as well as in four-day-old LPS-stimulated embryos. The expression of downstream genes was higher in *sting^(−/−)^* than in healthy WT fish. The late response was observed in *sting^(−/−)^* larvae following LPS treatment, demonstrating the importance of Sting-induced immunity during bacterial infection by activating the cGAS–STING pathway. Furthermore, adult *sting^(−/−)^* fish had a high mortality rate and significantly downregulated cGAS–STING pathway-related genes during *Edwardsiella piscicida* (*E. piscicida*) infection. In addition, we assessed NF-κB pathway genes following *E. piscicida* infection. Our results show fluctuating patterns of *interleukin-6* (*il6)* and *tumor necrosis factor-α* (*tnfα)* expression, which is likely due to the influence of other NF-κB pathway-related immune genes. In summary, this study demonstrates the important role of Sting against bacterial infection.

## 1. Introduction

The innate immune system recognizes invading pathogens through pattern recognition receptors (PRRs) [1]. PRRs include retinoic-acid-inducible gene 1 (RIG-1) receptors (RLRs), nucleotide-binding oligomerization domain-like receptors (NLRs), toll-like receptors (TLRs), AIM2-like receptors (ALRs), and C-type lectin receptors (CLRs). Cell surface PRRs (TLRs and CLRs) are responsible for sensing pathogens from the extracellular environment, cytoplasmic PRRs (NLRs, RLRs, and ALRs) are responsible for detecting intracellular pathogens, and endosomal PRRs (TLRs) are used to detect microbes that have entered phagolysosomes [2]. PRRs regulate the expression of inflammatory mediators through intracellular signaling cascades to eliminate pathogens from infected cells [3]. The RLR family plays an important role in recognizing non-self-signatures (of viral RNA) in the cytoplasm, and their signaling controls viral infection. RLRs modulate the host intracellular immune response by regulating the expression of interferons (IFNs) and antiviral genes and activating downstream transcription factors [4,5].

In addition to viral RNA, cytoplasmic DNA can induce innate immune defenses. Cytosolic DNA is a signal of pathogen entry or nuclear self-DNA leakage following DNA damage. The host cyclic GMP–AMP synthase (cGAS) senses cytosolic double-stranded DNA (dsDNA) and forms cyclic dinucleotides (CDNs). The binding of CDNs to stimulator of interferon genes (STING), an endoplasmic reticulum (ER)-associated sensor, initiates STING trafficking from the ER to the Golgi complex. Trafficking of the STING complex leads to recruitment and activation of TANK-binding kinase 1 (TBK1) and interferon regulatory factor 3 (IRF3) to induce IFN-1 and inflammatory cytokine production [6,7,8]. Furthermore, STING activation stimulates nuclear factor kappa B (NF-κB) and induces proinflammatory cytokine secretion [9].

The functional form of STING is a dimeric polymer containing monomers that arrange side-to-side and form a ligand-binding pocket [10]. The proposed molecular model based on biochemical and structural studies of STING has explained its active and inactive states. In the inactive state, the STING dimer on the endoplasmic reticulum is free and flexible, with a sequestered C-terminal tail. In this state, the reactive cysteine is uncoupled, and the linker regions in the STING dimer are crossed. When activated by 2′3′-cyclic GMP–AMP (cGAMP) binding, the dimeric angle of STING becomes closed, and a binding pocket is formed by a β-lid. In the active state, the sequestered C-terminal tail of STING is released, and the linker regions become uncrossed. The polymerized STING provides a scaffold for TBK1 by arraying its C-terminal tail and recruiting IRF3 for downstream signal propagation [11]. Investigation of the molecular interactions involved found similar mechanisms of activation for IRF7 and IRF3 by mitochondrial antiviral signaling protein (MAVS) and STING, respectively. However, IRF7 activation requires two phosphorylation events, whereas IRF3 activation only requires one; therefore, the regulation of IRF7 appears to be tightly controlled [12].

Current drug discovery efforts have focused on the cGAS–STING pathway and have identified agonists as antivaccine adjuvants or immunostimulatory agents in cancer [13,14]. Studies on zebrafish larvae indicated that Sting silencing markedly weakens the antiviral response against DNA viruses such as herpes simplex virus 1 [15]. EPC cell lines overexpressing zebrafish Sting and Mavs demonstrated the potential role of Sting in RNA and DNA virus-mediated teleost immunity against RNA and DNA viruses, and supported the hypothesis that Sting is largely conserved in vertebrates [16]. Reports on mucosal immunity provided by the cGAS–STING pathway provide experimental evidence for the role of this pathway in Ifn-mediated immunity in teleost fish [17]. Furthermore, this pathway plays a dual role during bacterial infection. It is involved in host defenses against bacteria while also promoting bacterial replication and survival [18,19,20].

Studies over the past few years elucidated the molecular mechanisms of Sting and its role in microbial detection and elimination [6,20,21,22]. Zebrafish are a vertebrate model organism for studying human diseases and understanding immunity [23,24]. Although some studies evaluated the antiviral and antibacterial immune activity of zebrafish Sting [15,16,17], the immune functions of Sting against pathogenic infection in fish have been poorly investigated. Moreover, in vivo studies are imperative for drawing conclusions concerning the microbial immunity in the presence of Sting.

In this study, we analyzed *sting* expression in early developmental stages and the immune-related organs of adult zebrafish. We established CRISPR/Cas9-mediated *sting* knockout zebrafish (*sting^(−/−)^*) and assessed the transcription of *sting* by comparing downstream gene expression between wild-type (WT) and *sting^(−/−)^* zebrafish. Furthermore, the roles of Sting in zebrafish survival and downstream gene modulation during bacterial infection were compared upon LPS stimulation of zebrafish larvae and *Edwardsiella piscicida* (*E. piscicida*) challenge in juvenile *sting^(−/−)^* and WT zebrafish. The data from our study may encourage future studies on the antibacterial activity of zebrafish Sting, and the *sting^(−/−)^* zebrafish model can be an important tool for further studies on the role of Sting in the zebrafish immunity.

## 2. Methodology

### 2.1. Animals

Wild-type (WT; AB strain) zebrafish were maintained as previously described [25]. Laboratory water circulation and filtration systems for the zebrafish cultures were maintained at a constant temperature of 28 ± 0.5 °C in a 14:10 h light–dark cycle. All animal experiments were approved by the Animal Care and Use Committee of Jeju National University.

### 2.2. CRISPR/Cas9-Mediated Generation of Sting^(−/−)^ Zebrafish

The *sting^(−/−)^* zebrafish was generated using CRISPR/Cas9 gene-editing technology [15]. The CRISPR/Cas9 *sting*-targeting site (5′ AGAGCGCGCAGCAGGCTGCC 3′) were designed using an online tool [http://zifit.partners.org/ZiFiT/ (accessed on 3 September 2019)]. Single-guide RNA (sgRNA) synthesis, microinjection, and mutant confirmation were performed as previously described [25]. The sgRNA target site was chosen to be 20-bp before the protospacer adjacent motif (PAM) sequence in exon 3 (Figure 1A). To confirm target site mutations in the genomic DNA, polymerase chain reaction (PCR) was carried out using T7E1 primers designed from intron 2 to exon 3 of the *sting* sequence after thatT7E1 assaywas performed (Table 1).

### 2.3. Sample Collection for RNA Extraction

#### 2.3.1. Tissue Collection from Adult Zebrafish

To evaluate the tissue-specific *sting* transcriptional patterns, healthy male and female zebrafish (5 per sex) were dissected after anesthesia, and tissues (including the brain, gill, heart, intestine, kidney, liver, muscle, ovary, spleen, and testis) were collected. Internal organs (including the intestine, kidney, liver, and spleen) and muscle were collected from fish challenged with bacteria. Immediately after isolation, all tissues were snap-frozen in liquid nitrogen and stored at −80 °C until RNA extraction.

#### 2.3.2. Embryo Sample Collection

To analyze *sting* expression during early development, zebrafish embryos at each developmental stage were collected based on morphological criteria [26]. A total of 50 embryos each of WT and *sting^(−/−)^* zebrafish were collected seven days post-fertilization (dpf) for downstream transcriptional studies and mutation analysis. Embryos were washed with RNA-grade 1 × phosphate-buffered saline (PBS) and stored at −80 °C for RNA extraction.

### 2.4. LPS Stimulation of Sting^(−/−)^ and WT Zebrafish Larvae

WT and *sting^(−/−)^* zebrafish larvae (3 dpf) were divided into six groups, each containing 20 larvae. Three groups were treated with 100 µg/mL LPS (*Escherichia coli* 0111:B4) and three were maintained as a PBS-treated control. Embryos were collected 6, 12, and 24 h after LPS treatment. RNA extraction and cDNA synthesis were performed as described in Section 2.6. Using quantitative real-time PCR (qPCR), the expression of downstream genes, such as *interferon phi 1* (*ifnphi1*), *caspase b*, *tnfα*, and *interleukin 6* (*il6*), in WT and *sting^(−/−)^* zebrafish was examined at various time points (qPCR). The same experiment was repeated, and larval survival was monitored for three days following LPS treatment.

### 2.5. Bacterial Challenge against Sting^(−/−)^ and WT Zebrafish

Two-month-old juvenile WT and *sting^(−/−)^* zebrafish were challenged with *E. piscicida* by the immersion method as previously described, with some modifications [27]. To examine the percentage mortality of *sting^(−/−)^* and WT zebrafish following *E. piscicida* infection, the zebrafish were divided into three groups of 20 individuals each. Two groups were wounded by removing ten scales before bacterial or PBS exposure, and the third group was used as a non-injured control. The first group was exposed to *E. piscicida* at a final concentration of 10^8^ CFU/mL, and the control groups were treated with the same volume of PBS in a total volume of 300 mL. After a 5 h immersion bath, the zebrafish were transferred to a new tank with 2 L of water, and half of the water was replaced with new water every day. Zebrafish were maintained within an incubator at 28 ℃ for 12 days to observe mortality. Feeding was paused one day before infection, and resumed two days post-infection (dpi).

For the challenge experiment, WT and *sting^(−/−)^* zebrafish were divided into three groups, each containing 24 zebrafish, and a wound was generated in two groups before bacterial treatment. These two groups (one from both WT and *sting^(−/−)^*) were exposed to *E. piscicida* at the final concentration of 10^8^ CFU/mL in a total volume of 300 mL for 5 h, and the other was maintained as a PBS-treated control without injury. The bath immersion and maintenance of zebrafish were performed as described in the mortality experiment. In addition, the zebrafish were not fed after bath immersion. Tissue samples, including the internal organs and muscles, at the infected site of six zebrafish from each group were collected at 6, 24, 48, and 72 h post-infection (hpi), as shown in Appendix A. Immediately after collection, tissue samples were snap-frozen in liquid nitrogen. RNA extraction and cDNA synthesis were performed as described in Section 2.6. The expression of downstream genes, including *tbk1*, *irf3*, *irf7*, *nf-κb*, *tnfα*, *ifnphi1*, *il6*, and an internal control gene *elongation factor-1α* (*ef1*α) in WT and *sting^(−/−)^* zebrafish were analyzed by qPCR at different time points post-*E. piscicida* challenge. Relative expression of the target genes was calculated with respect to *ef1*α expression using the Livak method [28,29]. The expression of target genes in *E. piscicida*-infected samples were normalized with respect to the PBS-treated controls.

To confirm that *E. piscicida* infection occurred, six randomly selected zebrafish from the *E. piscicida*-infected WT and *sting^(−/−)^* groups were sampled at 6 hpi. The genomic DNA was extracted, and the presence of *E. piscicida* was confirmed by PCR using *E. piscicida* and *gapdh* detection primers following agarose gel electrophoresis.

### 2.6. RNA Extraction and Complementary DNA Synthesis

RNA was extracted from zebrafish tissues and embryos using TRIzol reagent (Sigma-Aldrich, St. Louis, MO, USA). RNA concentrations were assessed using a Multiskan™ GO Microplate Spectrophotometer (Thermo Fisher Scientific, Waltham, MA, USA) at 260 nm. The quality of the RNA samples was confirmed by agarose gel electrophoresis. First-strand cDNA was synthesized from 3 μg RNA using a Prime Script™ 1st strand cDNA synthesis kit (Takara, Shimogyo-ku, Kyoto, Japan). The cDNA samples were diluted 30-fold and stored at −20 °C for PCR and qPCR analysis.

### 2.7. Transcriptional Analysis Using qPCR

The qPCR reaction mixture (10 µL) was prepared using 5 μL 2× TaKaRa Ex Tag SYBR premix, 0.5 μL each of the forward and reverse primers (Table 1), and 4 μL cDNA. qPCR reactions were performed using a Thermal Cycler Dice Real-Time System III (TaKaRa, Japan) under the following thermal conditions: initial denaturation at 95 °C for 10 s; 45 PCR cycles, each consisting of denaturation at 95 °C for 5 s, annealing at 58 °C for 10 s, and extension at 72 °C for 20 s, and finally, one melting cycle at 95 °C for 15 s, 60 °C for 30 s, and 95 °C for 15 s. Relative mRNA expression was calculated using the 2^−ΔΔCt^ method.

### 2.8. Statistical Analyses

All experiments were carried out in triplicate, and data expressed as the means ± standard deviation (SD). Student’s *t*-tests were used for statistical analysis, and *p* < 0.01 (indicated by *) and <0.001 (indicated by **) were considered statistically significant.

## 3. Results and Discussion

### 3.1. Expression Analysis of Sting

*Sting* and *β-actin* transcription during the embryonic developmental stages from the 2-cell stage to 7 dpf was analyzed using PCR, and the gel electrophoresis images are shown in Appendix A. Moderate *sting* expression is observed starting from the shield stage, and the *sting* expression pattern in our study is similar to that observed in previous studies [17].

*Sting* expression is observed in all analyzed tissues of the adult zebrafish, and the results are plotted as fold-values of the expression in muscle (Appendix A). High expression is observed in the kidney, gill, and testis, followed by in the spleen and brain. A previous study found that *sting* expression was highest in the kidney, followed by the gill, heart, and spleen [17]. We also examined the transcription of *stings* in the ovary and testis. Our results, in accordance with previous reports, show high expression of *sting* in important immune organs in fish, such as the kidney, gill, and spleen, which may be due to the role of Sting in autoimmunity [8]. However, they are slightly different regarding *sting* expression in the heart, muscle, and intestine. Adaptations occur in organisms depending on environmental and nutritional changes, and this may explain the different gene expression patterns observed in different studies [30,31,32]. Furthermore, moderate *sting* expression is observed in the brain, possibly due to Sting’s role(s) in controlling neuronal gene expression [33]. The transcription of *sting* in mandarin fish was highest in the gill, followed by the kidney, spleen, and blood [34].

### 3.2. Generation of Sting^(−/−)^ Zebrafish by CRISPR/Cas9 Gene Editing

The target site mutation with a 4-bp deletion was selected using nucleotide sequence analysis, as it produced a prematurely truncated protein (Figure 1B,C). The selected mutation was verified by PCR in WT and *sting^(−/−)^* larvae at 7 dpf using the original target site sequence as a primer (Figure 1D). The PCR-amplified band was observed in WT fish only, confirming the absence of *sting* in the *sting ^(−/−)^* group. The *sting^(−/−)^* zebrafish were normal in terms of survival, morphology, and fertility. *Sting* knockout mice have also been found to be viable and fertile [35].

### 3.3. Downstream Gene Expression Analysis in Sting^(−/−)^ Zebrafish

*Sting*-deficiency-mediated variations in the expression of downstream genes were analyzed using cDNA from 7 dpf WT and *sting^(−/−)^* zebrafish. The expression of downstream genes was normalized to that of *β-actin* (Figure 2), and we found that *tbk1*, *irf3*, *irf7*, and *nf-κb* were significantly upregulated in *sting^(−/−)^* zebrafish when compared to WT individuals. In addition to its role in innate immunity, STING regulates genotoxic stress homeostasis following pathway activation [36,37]. Knockdown of cGAS, STING, TBK1, and IRF3 in HeLa cells results in increased levels of micronuclei formation and chromosomal mis-segregation [38]. RLR, cGAS-STING, and non-RLR DExD/H-box RNA helicase pathways detect cytoplasmic nucleic acids and activate type 1 IFN and pro-inflammatory cytokines [5]. Deficiency in Sting can lead to chromosomal instability through micronuclei formation, which may explain the increased levels of *tbk1*, *irf3*, *irf7*, and *nf-κb* expression in *sting^(−/−)^* zebrafish. These observed transcription patterns support the involvement of Sting in regulating downstream gene transcription in zebrafish [39,40].

### 3.4. LPS-Induced Expression Modulation in Sting^(−/−)^ and WT Larvae

LPS is a major component of the outer membranes of Gram-negative bacteria and initiates the innate immune response of host organisms (and activates downstream pathways) following its recognition. Normally, LPS recognition and signal initiation are carried out by the TLR4 receptor [41]. Interestingly, zebrafish TLR4 has been identified as a negative regulator of TLR signaling and causes sequestration of TLR adaptors to inhibit the activation of NF-κB by MyD88 [42]. Recent studies observed LPS-induced activation of the cGAS–STING pathway and promotion of endometritis [43]. To investigate the role of zebrafish Sting in LPS recognition and innate immune response activation, transcriptional modulation of downstream genes (including *ifnphi1*, *caspb*, *tnfα*, and *il6*) was compared between WT and *sting^(−/−)^* zebrafish larvae at 6, 12, and 24 h post-LPS treatment (Figure 3). At 6 h post-LPS treatment, *ifnphi1* expression is significantly upregulated in both WT and *sting^(−/−)^* fish, but is more prominent in WT fish. At 12 h, the transcription levels of all the analyzed downstream genes are significantly upregulated in WT, while only *il6* is upregulated in *sting^(−/−)^* larvae. At 24 h, *ifnphi1* is downregulated in WT, while all the analyzed genes are upregulated in *sting^(−/−)^* larvae. Studies suggested the crucial role of STING upon LPS stimulation in inducing IFN production via cGAMP-primed enhancement [44]. LPS challenge in *Oplegnathus fasciatus* also caused a gradual increase in *ifn1* transcription at early time points, which reduced at later time points. Peak transcript levels were observed at 12 h in the blood and 24 h in the head kidney [45]. Our results (upregulated *ifnphi1* expression at 6 and 12 h, and downregulated expression at 24 h in WT) agree with this previous study. Furthermore, the observed fold reduction in *ifnphi1* upregulation in *sting^(−/−)^* compared to WT fish at early time points (6 and 12 h) indicates the involvement of a STING-mediated pathway in Ifnphi1 activation upon LPS recognition. In endothelial cells, mitochondrial DNA (mtDNA) is released from the mitochondria into the cytosol via mitochondrial pores induced by activation of the pore-forming protein gasdermin D. Gasdermin D is activated by LPS [46]. Subsequently, cytosolic mtDNA stimulates the cGAS–STING pathway. Another critical factor of Gram-negative bacterial sensing is the noncanonical inflammasome. In zebrafish fibroblasts, Caspase b binds to LPS directly via its N-terminal pyrin death domain, and its oligomerization is critical for pyroptosis [47]. In a previous study, *caspase* expression patterns were examined after LPS treatment to analyze the activation of noncanonical inflammasomes in zebrafish. It was revealed that Caspase a promotes pyroptosis canonically, similar to mammalian Caspase 1, while Caspase b induces non-canonical pyroptosis [48]. Moreover, the upregulated expression of downstream genes in *sting^(−/−)^* (but not in WT) fish at 24 h indicates that WT larvae recover from LPS-induced stress earlier than *sting^(−/−)^* larvae. Altogether, the lack of transcriptional stimulation of downstream genes in *sting^(−/−)^* zebrafish compared to WT zebrafish at early time points (6 and 12 h) after LPS treatment, and the late (24 h) response in *sting^(−/−)^* fish indicate the essential role of zebrafish Sting in conferring antibacterial immunity. Furthermore, LPS-treated WT and *sting^(−/−)^* larvae show 100% survival for three days. The results from the LPS tolerance study in zebrafish larvae 2, 5, and 10 dpf also show that mortality starts to occur at LPS concentrations of 150 µg/mL [49]. Therefore, the LPS concentration (100 µg/mL) used in this study was not lethal to WT or *sting^(−/−)^* zebrafish larvae.

### 3.5. Effects of Sting Deficiency on Susceptibility to E. piscicida Infection in Zebrafish

To study the effects of *sting* deficiency during bacterial infection, percent mortality was analyzed in *sting^(−/−)^* and WT zebrafish following *E. piscicida* infection (Figure 4A). Mortality begins in *sting^(−/−)^* and WT zebrafish at 4 and 5 dpi, respectively. Mortality in the *sting^(−/−)^* group reaches 55% at 8 dpi and remains unaltered thereafter, whereas it reaches 10% in WT fish at 8 dpi and remains unaltered. CDN-mediated activation of STING leads to the activation of genes that control pathogen replication and boost host adaptive immunity [8,50,51,52]. The involvement of the cGAS–STING pathway in Gram-positive and Gram-negative bacterial infection has been reported previously. However, the role of the cGAS pathway in bacterial infection is complex compared to its role in antiviral responses [6]. The importance of Sting in the antibacterial immunity of zebrafish is clearly demonstrated by our observed percent mortality values.

### 3.6. Temporal Expression Analysis in Zebrafish upon E. piscicida Infection

To understand the role of Sting in controlling immune-related gene expression during bacterial infection, we compared the modulation of downstream gene expression in *sting^(−/−)^* and WT zebrafish following *E. piscicida* infection. The relative mRNA expression of *tbk1* is significantly downregulated in the *sting^(−/−)^* group at 24 and 72 hpi, while no significant changes are observed in the WT control group (Figure 4B). The mRNA expression of *nf-κb* is significantly upregulated at 6 and 48 hpi in the WT control group, and significantly downregulated at 6 and 72 hpi in the *sting^(−/−)^* group (Figure 4C). The transcription of *irf3* is significantly upregulated at 48 hpi in the WT control group; however, no significant change is observed in the *sting^(−/−)^* group (Figure 4D). The transcription of *irf7* is upregulated at 6 hpi in the WT control group, but is significantly downregulated at 6 and 24 hpi, and upregulated at 72 hpi in the *sting^(−/−)^* group (Figure 4E). The transcription of *ifnphi1* is significantly upregulated at 6 hpi in the WT control group and significantly downregulated at 48 hpi in both the WT and *sting^(−/−)^* groups (Figure 4F). *tnfα* transcription is significantly upregulated at 6 hpi in the WT control group, whereas it fluctuates in the *sting^(−/−)^* group (Figure 4G). The transcriptional pattern of *il6* is similar to that of *tnfα* in both groups (Figure 4H), which suggests that these two genes may be regulated by the same mechanism.

Studies using murine models elucidated a STING-mediated mechanism that contributes to NF-κB activation [9]. STING-mediated activation of IRF3 and NF-κB is species-dependent. Observations using human and mouse primary immune cells indicate strong IRF3 and weak NF-κB responses to STING alleles (reporter signaling, approximately 40–60-fold and 15-fold, respectively) [8,51,53]. The activation of *sting* alleles from *Danio rerio* (zebrafish) and *Salmo salar* (salmon) in human cells demonstrated a stronger response by NF-κB compared to that by IRF3 (>100-fold stimulation). Furthermore, the CTT motif of zebrafish Sting dramatically enhances NF-κB signaling by recruiting Traf6 (Tnf receptor associated factor 6) [39]. The observed changes in *nf-κb* and *irf3* expression post *E. piscicida* challenge in the current study supports these previous results, as *sting*-deficient zebrafish show decreased expression of *nf-κb* and *irf3*. Since cytokines play a critical role in host immune defense and repair mechanisms, we analyzed the transcriptional modulation of cytokines *il6* and *tnfα* upon *E. piscicida* challenge [54]. Transcription modulation of *il6* and *tnfα* at an early time point (6 hpi) is similar to that of *nf-κb* and *irf7*. The expression patterns at 6 hpi clearly indicate that *sting* deficiency inhibits *ifnphi1*, *il6*, and *tnfα* expression through the *nf-κb* pathway. However, different cellular signaling pathways may be involved in modulating *il6* and *tnfα* expression at later time points (24, 48, and 72 hpi). Previous studies reported that deficiencies in TBK1 induced TNF*α*-mediated cell death [55,56]. This may explain the upregulation of *tnfa* and *il6* in *sting^(−/−)^* zebrafish at 24 and 72 hpi, as *tbk1* is downregulated at these time points. Further, mortality starts in *sting^(−/−)^* fish at 4 dpi, while it begins at 5 dpi in WT fish. Therefore, *tnfα* and *il6* upregulation at 72 hpi may result from early death signaling in *sting^(−/−)^* zebrafish. LPS-stimulated ROS generation in cardiomyocytes induces the translocation of NLRP3 from the nucleus to the cytoplasm via a STING-independent pathway [57]. The presence of NLRP3 inflammasomes is critical in triggering the expression of proinflammatory cytokines [58]. ROS levels in the immune organs of fish increase upon bacterial infection, which may trigger translocation of NLRP3 to the cytoplasm in a Sting-independent manner, inducing inflammation and apoptosis [59,60]. This may also explain the upregulation of *tnfα* and *il6* at later time points in the *sting^(−/−)^* group. STING activation by *L. monocytogenes* triggers *ifn1* to downregulate cell-mediated immunity [61]. IFN-1 is mostly related to antiviral immunity; however, it plays a role in antibacterial responses in mice [62,63]. A study on zebrafish Ifnphi1 and Ifnphi3 activation by Irf1, Irf3, and Irf7 shows that Irf3 acts both as a positive and negative regulator of *ifn* genes, depending on Irf1, Irf3, and Irf7 load in the cell [64]. This might explain the observed differential expression patterns of *irf3*, *irf7*, and *ifnphi1*.

## 4. Conclusions

In summary, we established a *sting^(−/−)^* zebrafish model to analyze the role of zebrafish Sting against bacterial infection. The transcription of genes downstream of *sting* (*tbk1*, *nf-κb*, *irf3*, and, *irf7*) was compared in WT and *sting^(−/−)^* zebrafish. The elevated expression of downstream genes in the *sting^(−/−)^* zebrafish illustrates the role of zebrafish Sting in controlling downstream gene expression. Furthermore, the role of Sting in survival and downstream gene modulation during bacterial infection were studied via *E. piscicida* challenge in *sting^(−/−)^* and WT zebrafish. The observed premature mortality in *sting^(−/−)^* zebrafish compared to WT zebrafish, and the differential expression patterns of downstream genes observed during *E. piscicida* and LPS stimulation, suggest that zebrafish Sting plays an essential role against bacterial infection by controlling downstream gene expression. The results of our study provide a reference for future studies on the antibacterial and other immune responses of zebrafish. Further, the understanding of the molecular pathways behind human diseases is being advanced by disease modeling in zebrafish. The *sting^(−/−)^* zebrafish model should be a key component of future research on antibacterial activity in humans based on these findings.

## Figures and Tables

**Figure 1 biomolecules-13-00324-f001:**
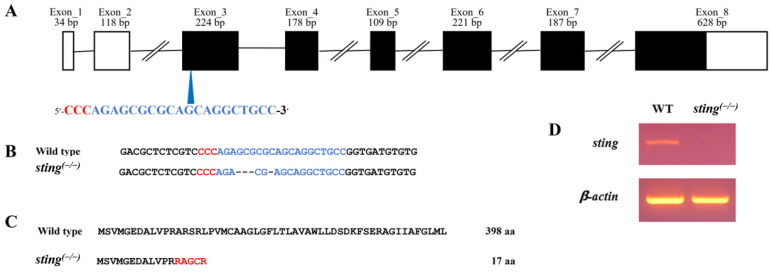
Generation of *sting^(−/−)^* zebrafish. (**A**) Schematic illustration of organization of zebrafish *sting*. The white and black boxes represent untranslated regions and open reading frames, respectively. The blue arrowhead indicates the single guide RNA (sgRNA) target site. The PAM and the sgRNA target sequences are indicated in red and blue letters, respectively. (**B**) The obtained mutation in *sting^(−/−)^* fish using CRISPR/Cas9 gene editing. Deleted nucleotides in the *sting^(−/−)^* genome are indicated with hyphens. (**C**) Amino acid prediction of Sting in the *sting^(−/− )^* and WT zebrafish genomes. The altered amino acids are represented with red letters. (**D**) Confirmation of mutation via RT–PCR using *sting^(−/−)^* mutation-specific primers in 7 dpf larvae of *sting^(−/−)^* and WT zebrafish.

**Figure 2 biomolecules-13-00324-f002:**
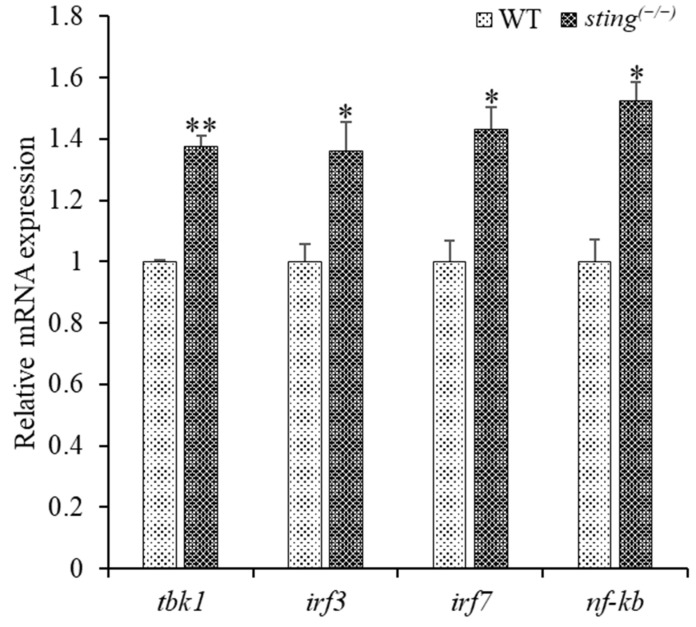
Comparison of transcription of downstream genes in 7 dpf larvae of *sting^(−/−)^* and WT zebrafish. The internal control gene, *β-actin*, was used to analyze the relative mRNA expression of genes downstream of *sting* using the Livak method. Experiments were performed in triplicate, and the error bar represents the standard deviation (SD). Student’s *t*-tests were used for statistical analysis, and *p* < 0.01 (indicated by *) and <0.001 (indicated by **) were considered statistically significant.

**Figure 3 biomolecules-13-00324-f003:**
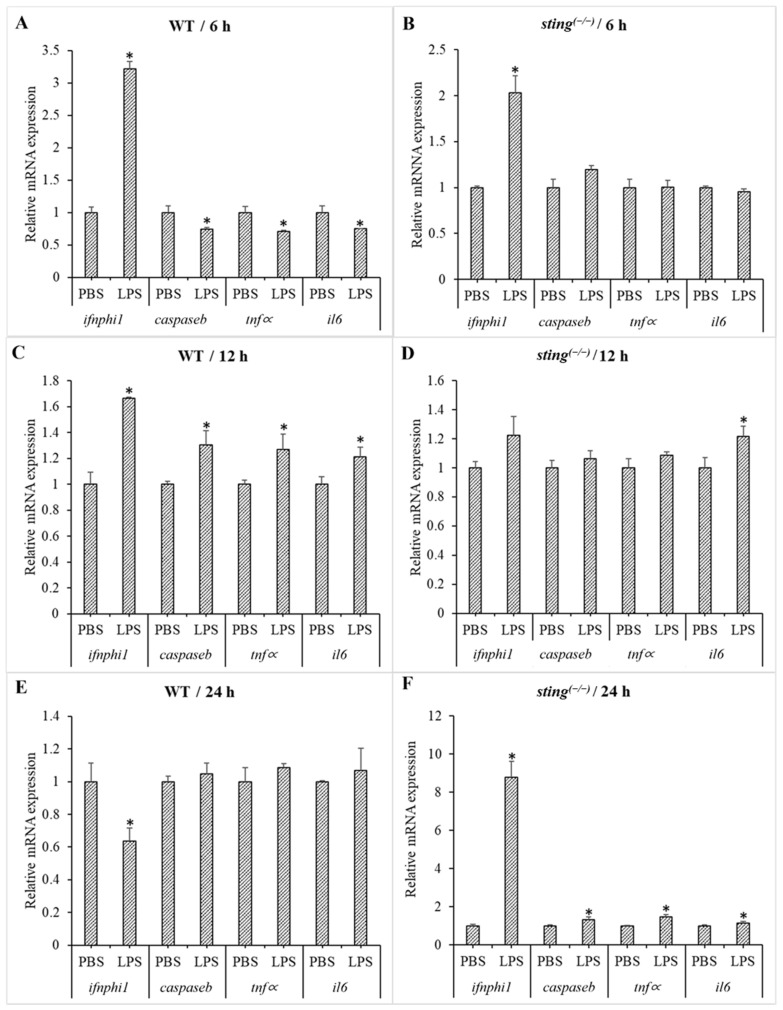
Comparison of gene expression in WT and *sting^(−/−)^* zebrafish larvae under LPS stimulation. The larvae samples were collected 6, 12, and 24 h post-LPS treatment of 3 dpf WT ((**A**,**C**,**E**) respectively) and *sting^(−/−)^* ((**B**,**D**,**F**) respectively) zebrafish larvae. *Ef1∝* was used as an internal control to analyze the relative mRNA expression of genes downstream of *sting* using the Livak method. Transcription levels of target genes in the PBS-treated group were considered as 1, and expression levels in the LPS-treated groups were normalized to those in the PBS-treated group and are represented as fold values. Standard deviation (SD; *n* = 3) is indicated by the error bars. Significantly differentially transcribed genes (when compared to the respective PBS-treated control) are marked with an asterisk (*: *p* < 0.05).

**Figure 4 biomolecules-13-00324-f004:**
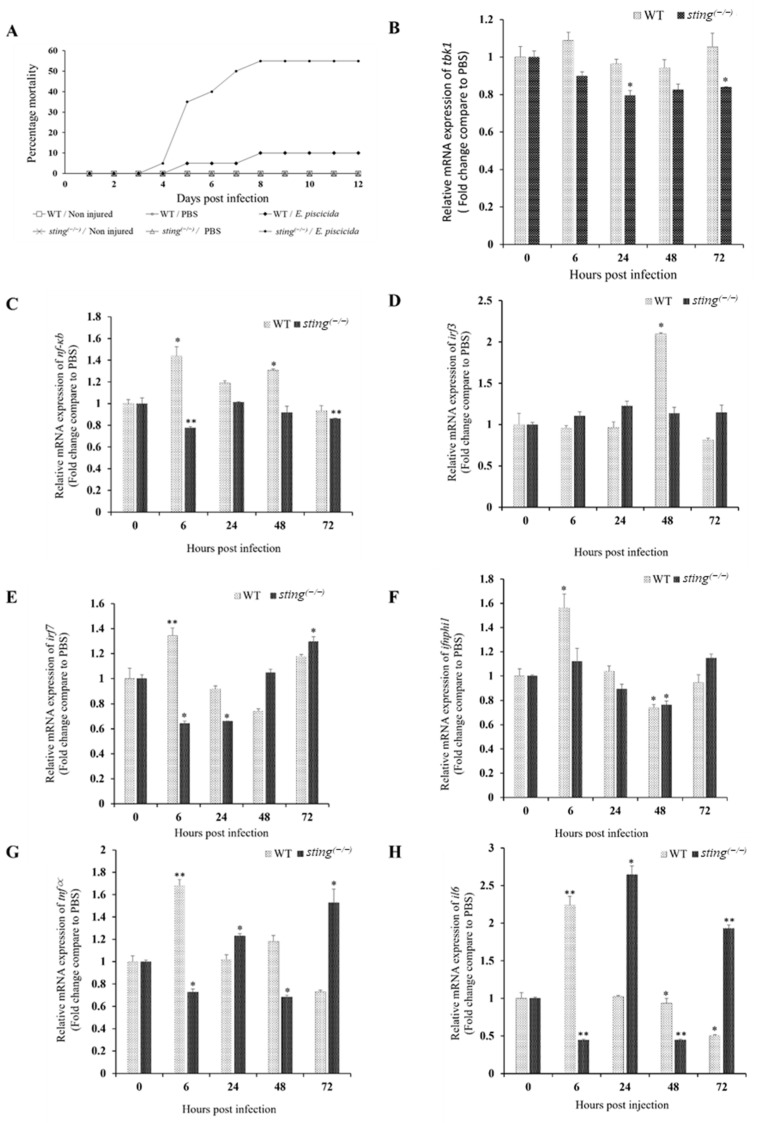
*E. piscicida* challenge in *sting^(−/−)^* and WT zebrafish. (**A**) Percent mortality of *sting^(−/−)^* and WT zebrafish post-*E. piscicida* infection. Two-month-old juvenile zebrafish were wounded, and *E. piscicida* infection was performed via bath immersion for 5 h. (**B**–**H**) Temporal transcription of *tbk1*, *nf-κb*, *irf3*, *irf7*, *ifnphi1*, *tnfα*, and *il6* in juvenile *sting^(−/−)^* and WT zebrafish upon *E. piscicida* infection. *ef-1α expression* was used as an internal control to analyze relative mRNA expression using the Livak method, and gene expression was normalized to that of the respective PBS control group. The error bars and the asterisk (* *p* < 0.01, ** *p* < 0.001) indicate the standard deviation and significantly different transcription levels compared to the respective control, respectively.

**Table 1 biomolecules-13-00324-t001:** The primers used in this study.

Gene	Application	Sequence (5′-3′)
*sting*	sgRNA F	TAATACGACTCACTATAGGCAGCCTGCTGCGCGCTCTGTTTTAGAGCTAGAAATAGC
	sgRNA R (Universal)	GATCCGCACCGACTCGGTGCCACTTTTTCAAGTTGATAACGGACTAGCCTTATTTTAACTTGCTATTTCTAG
	T7E1 F	GCAGTCATTTCTGTGTGGCTCTG
	T7E1 R	TCCAGCATGAGCCCAAAAGC
	qPCR F	TTTGGCGAGAGAGAACGGAAGC
	qPCR R	AGAGCTCTCTGGAGAGGTAATGAGG
	Mutation-specific F	TCGTCCCCAGAGCGCGCA
	Mutation-specific R	TCCGGCGCACATGAGGTTGAAGTGG
*tbk1*	qPCR F	CCTGTGGATGATGTCCGACC
	qPCR R	GGCGAACAGCTTGACGATG
*nf-κ* *b (p65)*	qPCR F	CAAAGATCTGGGAGGAGGAGTTCG
	qPCR R	GATCTTCAGCTCAGCAGTGTTAGGAG
*irf3*	qPCR F	CTGTACCTGATCACACTGCCATTCC
	qPCR R	GCCTGACTCATCCATGTTTCTGTGG
*irf7*	qPCR F	GAAGAGACCTTGGTGACGCG
	qPCR R	GAGACTGTGAAGTGCACATCGG
*tnfα*	qPCR F	CTCTCCGCTCTTCAGTTGACC
	qPCR R	GTGTGGTTTTGCCGTGGTC
*il6*	qPCR F	GATGAGGAGTACTTGCCGGG
	qPCR R	CCTGAGCCTAAATCCATGATCGC
*ifnphi1*	qPCR F	CCGCTTGTACACCTTGATGGAC
	qPCR R	GCCACACATTCTTTGAGGTCAG
*β-actin*	qPCR F	GCACATCTGCTGTAACAAGATCC
	qPCR R	GTCAGCAGATTCTGTCTGGC
*ef1* *∝*	qPCR F	CTCCTCTTGGTCGCTTTGCT
	qPCR R	CCGATTTTCTTCTCAACGCTCT
*caspb*	qPCR F	CAAGCAGAACGAACGTGCAAAGC
	qPCR R	TGCGAGATGATCTGCTGGATGG

## Data Availability

Not applicable.

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
