# Peer review of "CRISPR/Cas9-Induced Knockout of Sting Increases Susceptibility of Zebrafish to Bacterial Infection"

_biomolecules, 2023, doi:10.3390/biom13020324_

Round 1
Reviewer 1 Report
The authors investigated the role of Sting in bacterial infection using a CRISPR/Cas-mediated zebrafish knockout model to demonstrate the use of this animal model in studies of human immunity. However, a major concern of this reviewer is the overall applicability of this model. The authors adequately show the importance of Sting in zebrafish and the relevance of these studies to the understanding of zebrafish immunity but do not make the connection between these findings and human disease. Indeed, more than 26,000 protein-coding zebrafish genes show approximately 70% orthology to disease-causing human genes, such that these animals have become powerful models of human disease. Thus, it would be prudent for the authors to discuss the connection of these findings to increase the overall merit of this study. In addition, the English language and style could use a moderate amount of editing for grammar and fluency, spelling and formatting.
Moreover, there are a few minor points that should be addressed:
1 - Number the figures and tables consistently throughout the document and in the supplemental materials, and make sure the numbers in the in-text citations match (i.e., Table 1 vs. Table 01).
2 - Line 142 (Methodology): Describe how the dosage for LPS infection (100 ug/mL) was determined. Did you perform a dose curve experiment?
3 - Line 147 (Methodology): For full disclosure, add a brief description of your version of the immersion methods that includes the modifications from the original version.
4 - Line 151 (Methodology): Please describe how the final concentration for bacterial infection was determined?
5 - (Methodology): Please provide complete information for the manufacturer citations (i.e., location should include city, state and country).
6 - Line 186 (Results and Discussion): i) "images of the gel electrophoresis" are shown in Suppl. Fig. 1B not 1A. ii) Also, It is not useful to show only a few MW markers without providing information or any context to let the reader know the size. Either provide this information or only show the bands for sting expression vs. those for b-actin.
7 - Lines 189-190 (Results and Discussion): The results described here are shown in Suppl. Fig. 1A not 1B.
8 - Line 240 (Results and Discussion): The results would be more convincing if you conducted the LPS tolerance test to determine the precise dosage to use in the present study.
Reviewer 2 Report
In this manuscript, the authors demonstrate in vivo the protective role of STING against bacterial infection using zebrafish as a model. The results obtained here confirmed what has been previously reported in mice.
In Figure 3, I do not really understand the use of LPS since it is not an activator of the cGAS-STING pathway per se. I agree that in murine endothelial cells, sensing of cytosolic LPS activates the non-canonical inflammasome leading to a putative mtDNA release that may be sensed by cGAS (PMID: 32164878) and that this non-canonical inflammasome activation by cytosolic LPS seems to be conserved in zebrafish (PMID: 30076291). LPS is well known to promote NF-kB activation (through TLR4 stimulation in mammals) leading to the production of pro-inflammatory cytokines such as IL-6 and TNFa but here, the production is rather null after stimulation of zebrafish larvae with LPS. Maybe 100 ug/ml was too slow. Given that at the used dose, LPS did not seem to stimulate the TLR4, I do not think that the LPS could then trigger the non-canonical inflammasome activation.
LPS stimulation also promotes type I IFN production as a consequence of TBK1 and IRF3/7 activation. TBK1 and IRF3/7 are activated in a post-transcriptionally manner through phosphorylation, I therefore do not understand why their expression levels have been explored in figure 3. Investigation of type I IFN expression through qPCR would have been more judicious as in Figure 4.
Figure 4B, C, D and 4. mRNA expression of TBK1, NF-kB, IRF3 and IRF7 is not informative since as mentioned above, these proteins/factors are activated through phosphorylation but not through regulation of their expression. Analyses 6 hrs post-infection clearly show that the lack of STING prevents the production of IFN, TNFa and IL-6 which may explain why STING ko fishes are more sensitive to infection. Nevertheless, how do the authors explain that at longer time post-infection, the lack of STING does not significantly inhibit the production of those cytokines, the fluctuations are then puzzling?
Other comments:
What kind of NF-kB subunit was investigated by qPCR this study? p65, p50, another?
In Figure 2, how do the authors explain the increase in mRNA expression of TBK1, IRF3 and IRF7? Could it be a way to compensate the loss of STING?
Minor comments:
Line 186. The transcription of sting and êžµ-actin in embryonic developmental stages from 2-cell stage to 7 dpf was 185 analyzed by PCR, and the images of gel electrophoresis are shown in Supplementary Figure 1B, not 1A
Line 190. The sting expression was observed in all analyzed tissues of adult zebrafish, and the results are plotted as fold 189 values of the expression in muscle (Supplementary Figure. 1A not 1B)
Round 2
Reviewer 1 Report
The authors have done a good job of revising the manuscript thus far, and version 2 is highly improved. However, there are still more issues that need to be addressed. This reviewer offers the following:
1 - Another pass for English language editing is needed to further improve the grammar and fluency and convention issues throughout the document. For example: 1) Keywords: scientific notation (italics); 2) Line 108 (Methods): "sites"? How many sites were targeted? You only describe one. Please list the other sites or change this word to the singular form (i.e., "site"); 3) Line 122: Missing punctuation; 4) Line 174 (Methods): Please capitalize "Supplementary Figure 2A" and please correctly use the term "snap-frozen" NOT "snapped and frozen"; 5) Line 182 (Methods): Add a space between E. and piscicida; 6) Line 206 and many other places throughout the document: Keep the italics for the gene name, but if the word starts the sentence, then it should be capitalized; 7) Line 213: Do not pluralize the name of this gene (i.e., sting NOT stings); 8) Line 84: "But also" is part of a correlative conjunction pair (i.e., "not only... but also..."). Please revise this sentence to correct the issue with paralellism; 9) Line 294: Define the acronym "CDN"; 10) Line 329: Grammar is wrong. Should say "response by NF-kB compared to that by IRF3"; 11) Line 338: subject-verb agreement (i.e., deficiencies induce NOT induces). These examples are just a few. A native English language editor would help you to clean the document further. Please consider.
2 - Line 48: "PRRs of the cell surface..." This sentence is awkwardly written and misleading, as PRR recognition is the first step, followed by phagosome formation and phagolysosome maturation. Indeed, PRR recognition and binding initiates subsequent microbicidal and pro-inflammatory effects, such as phagocytosis. The way you have written this sentence implies that PRR recognition comes after microbes have already entered phagolysosomes. Please revise for clarity.
3 - Line 88: "Although some have evaluated..." What does the "some" refer to? Do you mean "some studies have evaluated"? Please revise for clarity.
4 - Line 97 and Line 362: "The data from our study..." Zebrafish research is important because it allows us to gain insights into the pathophysiology of human diseases. Thus, these data should encourage future studies on antibacterial activity in humans using the sting-/- zebrafish model as an important tool. This is why this study is significant to the research community at large. Please revise these sections to appropriately reflect the significance of this study.
5 - Line 113 (Methods): Please change the table reference to match the table label (i.e., Table 1 NOT Table 01).
6 - Fig. 1D: The MW marker lanes ARE STILL showing in this figure. Please either label the marker bands or crop them from the figure, as they offer no additional information in the way they currently appear.
7 - Line 141: Please remove redundant information (i.e., "after anesthesia was administered").
8 - Line 176 (Methods): You added "caspaseb" to Table 1 (list of primers) for qPCR. Please add this protein to the text here.
9 - Line 209: Zebrafish (Danio rerio) embryonic development was examined here. So, this comparison is not clear. Do you mean "was similar to that observed in previous studies? Please clarify.
10 - Line 239, 249 and elsewhere throughout the document: Please use correct formatting for protein/gene names (i.e., for fish, protein symbols are not italicized, and the first letter is uppercase (e.g., Sting).
11 - Line 269, 296 and elsewhere throughout the document: Please be consistent with capitalization conventions used throughout the document (i.e., i) Gram should be capitalized and never hyphenated when used as Gram stain; ii) gram negative and gram positive should be lowercase and iii) only hyphenated when used as a unit modifier). Based on this information, please use the following examples to edit the document accordingly. Examples:
i) Gram staining
ii) gram negative
iii) gram-positive bacteria
12 - Line 271: What were the results of this study (i.e., Ref 50)?
13 - Lines 286-288: What organism was the LPS tolerance study performed in (i.e., Ref 51)? Please add this information for clarity.
14 - There are more than a single reference cited. Please change the heading to "References".
Reviewer 2 Report
Referees' comments have been addressed in a satisfactory manner. I recommend acceptance for publication
Author Response
The reviewer has gone through the manuscript and mentioned to accept the applied manuscript.
Thank you for the revision.